# Influence of Thermal Regimes on the Relationship between Parasitic Load and Body Condition in European Sardine along the Catalan Coast

**Xènia Frigola-Tepe [1], Marta Caballero-Huertas [1], Jordi Viñas [2] and Marta Muñoz [1,\*]**

[1] Institute of Aquatic Ecology, Universitat de Girona, Maria Aurèlia Capmany 69, 17003 Girona, Spain

[2] Genetic Ichthyology Laboratory (LIG), Department of Biology, Universitat de Girona, Maria Aurèlia Capmany 69, 17003 Girona, Spain

\* Correspondence: marta.munyoz@udg.edu

**Abstract:** The small pelagic European sardine presents high commercial and ecological values. Due to its cold-temperate water affinity, stocks are affected by global warming. Water temperature rise may change primary productivity patterns, negatively affecting fish condition and increasing parasite incidence. In this context, sardine health status was evaluated through the annual cycle on the Catalan Coast using thermal regimes comparison. Morphogravimetric parameters, sex and gonadal stages were assessed; infection by nematodes was characterised, and body condition was estimated by the Le Cren Factor and lipid content measured using a fish fat meter. Significant statistical differences were observed in spawning dynamics, body condition, and parasite infection between thermal regimes. Sardines from the colder north area had better condition and an earlier spawning, with lower parasite incidence (in terms of total prevalence, mean intensity and abundance) than those from the southern coast. *Hysterothylacium* spp. was the most abundant nematode, while *Anisakis* spp. prevalence was null in the two locations. Seasonal differences in nematode load were observed along the Catalan Coast, with lower prevalence during the summer and higher in winter-spring. Although previous studies have underestimated parasite influence on sardine health status, parasite abundance and sardine condition were negatively correlated. Seawater temperature and primary productivity are the proposed factors promoting differentiation in nematode infection and fish condition throughout the annual cycle and between locations.

**Keywords:** *Anisakis*; health status; *Hysterothylacium*; fat content; nematode; NW Mediterranean; *Sardina pilchardus*

## 1. Introduction

The European sardine, *Sardina pilchardus* (Walbaum, 1792), is a cold-temperate water pelagic species of the Clupeidae family. This small planktivorous fish performs a key functional role in the marine ecosystem, exerting *wasp-waist control* (i.e., transferring energy from the lower to the upper trophic levels) [1,2]. It is distributed from the northeast of the Atlantic Ocean, the Sea of Marmara, the Black Sea and the Mediterranean Sea [3]. In those regions, sardine is one of the most important commercial fishery resources [4] because it is appreciated for its excellent gastronomic value in terms of taste and nutritional benefits [5,6].

This species has been defined as a capital breeder [7] since the reproductive cycle is strongly related to the annual somatic conditions. Therefore, sardines feed intensively during the maximum primary production in spring-summer and accumulate sufficient energetic reserves to assume the cost of reproduction during the most oligotrophic period in winter [8,9]. This trait is in accordance with the demonstrated close relationship

between reproductive potential and recruitment with fish bioenergetics, which depends on environmental factors [10,11].

A recent FAO report [12] emphasised that sardine biomass was below biologically sustainable levels in the Mediterranean. In this respect, the data indicate that sardine stocks are made up of younger individuals with smaller total length and sexual maturation size (L50) than in former years (before 2008) [13,14]. This reduction has been attributed to intensive fishing and water temperature increase [15,16], which can lead to a decrease in energy reserves and sardine health status [17]. All of these stressors could alter spawning dynamics and reduce fecundity and, in consequence, recruitment [18]. Among other factors that have been associated with this decline in condition are the changes in zooplanktonic composition, pollution, or the increase of parasites that may result from seawater temperature rise [19,20]. In this regard, even though few studies focus on the effect of parasitic load on fish condition, previous results have showed their impact on fish health status [21–23]. In this blend of biological and physicochemical variables, the need for a study considering their effects on sardine condition and reproduction and their interrelationship has emerged. In the NW Mediterranean, primary production is determined by the strong vertical mixing and contribution of fluvial nutrient loads [24]. Considerable differences in seawater temperature are reported in this area [25]. Therefore, due to its environmental heterogeneity and particular hydrographical feature, it is of primary interest to study the life history traits of sardine in this area, as well as its parasite infection.

In this context, the objective of the present study was to analyse European sardines' somatic condition throughout the complete reproductive cycle in two different thermal regime areas in the NW Mediterranean, specifically in the Catalan Coast. Notable studies on sardine status in the Mediterranean have focused on condition (e.g., [26,27]), reproduction (e.g., [28]) or parasite infection (e.g., [29]). In this context, our work expands on previous research, obtaining a general picture of the current health status of sardines in the NW Mediterranean by analysing condition and nematode infection (and their interrelationships) within the reproductive context.

## 2. Materials and Methods

### 2.1. Fish Sampling

A total of 1375 specimens of European sardine, *Sardina pilchardus*, were collected monthly from October 2019 to July 2021, as reported in Table 1. Samples were caught by commercial purse seiners in the following two areas with different thermal regimes in the Catalan Coast, located in the Northwestern Mediterranean Sea (Mediterranean Sea, GFCM—GSA 6): Costa Brava in the north, with lower water temperature, and the area close to Ebro Delta in the south [25]. Samples were obtained from fishing harbours located in the north (L'Escala, Palamós and Blanes) and in the south (Tarragona, Cambrils and Ametlla de Mar) (Figure 1).

**Table 1.** The number of fish sampled per thermal zona (northern and southern coast) in each season (autumn, winter, spring and summer).

|  | Autumn | Winter | Spring | Summer |
|---|---|---|---|---|
| **North** | 224 | 241 | 149 | 233 |
| **South** | 175 | 181 | 80 | 92 |

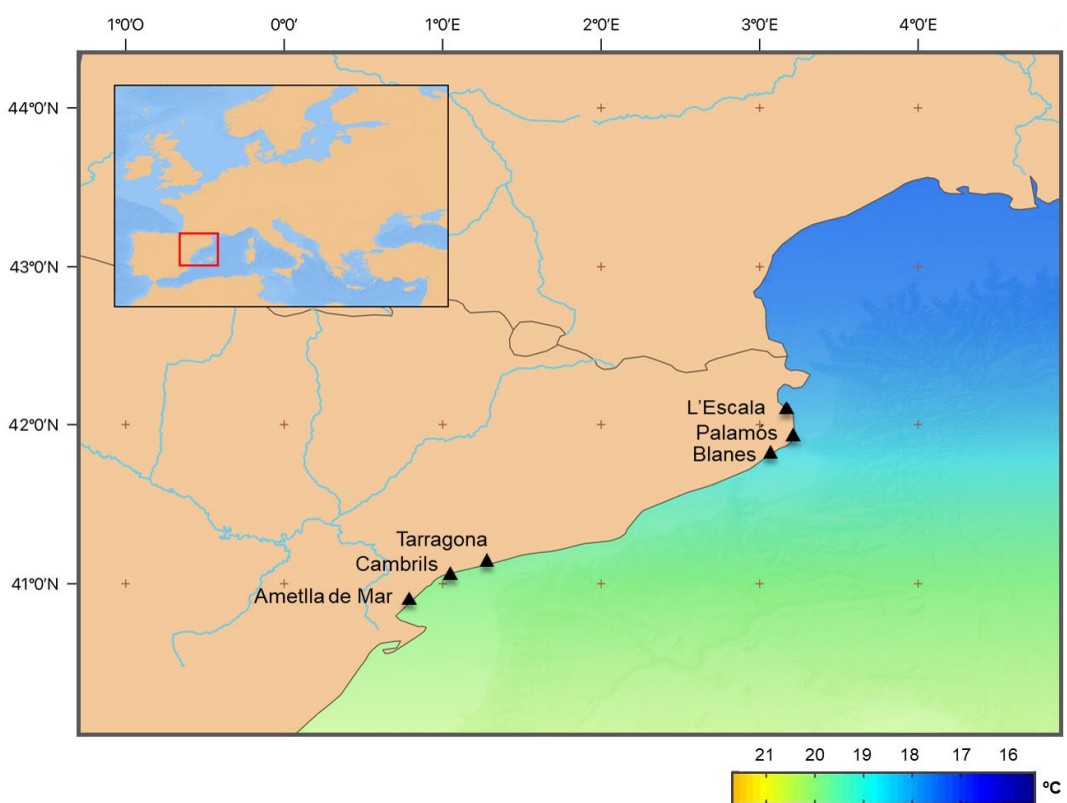

**Figure 1.** Map of the Catalan Coast in the western Mediterranean Sea showing sea surface temperature (SST, °C) to delimit thermal regimes and the fishing harbours where *Sardina pilchardus* was caught (FAO-GFCM geographical subarea GSA-06, Northern Spain). Along the northern coast, the locations of l'Escala, Palamós and Blanes are indicated; on the southern coast are the locations of Tarragona, Cambrils and Ametlla de Mar. Temperature (°C) was retrieved monthly from HadISST sea surface temperature time series (1900–2020) for the period 2019–2020 (https://www.metoffice.gov.uk/hadobs/hadisst/ (accessed on 22 February 2022 )).

In order to delimit thermal regimes, sea surface temperature (SST, °C) from the sampling areas was retrieved monthly from the HadISST sea surface temperature time series (1900–2020) for the period 2019–2020 (https://www.metoffice.gov.uk/hadobs/hadisst/ (accessed on 22 February 2022)). When comparing sampling areas, higher SST values were observed in the Ebro Delta area (19.83 °C) than on the northern coast (18.57 °C) ($p = 0.000$). In both areas, monthly SSTs started to increase from April to May, peaked in August, and decreased towards the coldest months of January to March.

All samples were obtained shortly after being landed, stored in ice and transported to the laboratory, where they were freshly processed. For each specimen, the total body length ($L_T$; ± 0.1 cm), total body mass ($M_T$; ± 0.01 g), eviscerated body mass ($M_E$; ± 0.01 g) and gonad mass ($M_G$; ± 0.1 mg) were recorded.

### 2.2. Reproductive Cycle

Sex was macroscopically determined, based on gonad shape and structure. Gonad developmental phases were classified into the following six gonad developmental stages as described in Brown-Peterson et al. [30]: immature (IM, sardine has not reached sexual maturity); regenerating (RT, mature but reproductively inactive); developing (DV, gonads increasing in size with gametes that are beginning to develop); spawning capable (SC, ready for the reproduction but sardine has not begun to spawn); actively spawning (AS, sardine is releasing eggs) and regressing (RG, gonads almost empty of gametes).

The energy allocated to reproduction was estimated by calculating the gonadosomatic index (GSI), which is the relationship between gonad mass ($M_G$) and eviscerated body mass ($M_E$). This index was calculated as GSI = $100 M_G M_E^{-1}$ [31].

### 2.3. Condition

Sardines' condition or energetic fitness was assessed according to the energy reserves of the fish, using morphogravimetric and bioenergetic indices. As stipulated by Van Beveren et al. [13], sardines follow an allometric growth pattern, allowing us to use the Le Cren condition factor [32] to define the individual fish condition [33].

$$K_n = \frac{M_T}{M_r} = \frac{M_T}{\alpha L_T{}^\beta}$$

$M_T$ is the body weight of a sardine individual, and $M_r$ represents the theoretical weight of a given total length predicted by a length–weight relationship. In this equation, $L_T$ is the total length, and $\alpha$ and $\beta$ are constants obtained by the averages of the regression line of the logarithms of length and mass. When $Kn$ equals 1, the sardine has a standard physical condition; when $Kn$ exceeds 1, the sardine has a higher than average physical condition; when $Kn$ is less than 1, the sardine manifests a lower than average condition.

Tissue fat and mesenteric fat content are considered good proxies for estimating individual fish condition [34] since they constitute the primary and secondary lipid repositories in sardines. Estimation of the tissue fat content was determined with an indirect bioenergetic index using a Distell Fish Fat Meter [35]. We used the MFM-992 fatmeter with a small microstrip sensor that allows the rapid measurement of the water content of the tissues and provides the relative fat content [34,36], which has been demonstrated to provide a very good correlation with total lipid content in sardine [37]. Tissue fat content was averaged by considering measurements from both sides of the specimen along the lateral line. Finally, and in addition to this information, a visual assessment of the mesenteric fat associated with the visceral organs was defined following the seven stages defined by Van der Lingen and Hutchings [38].

### 2.4. Parasitological Examination

All fresh samples of European sardine were examined for internal nematode presence. The entire viscera were removed from the body cavity, and the gills and internal organs (stomach, intestine, pyloric caeca, liver and gonads) were examined under a stereomicroscope (Zeiss binocular magnifier).

The detected nematodes were collected and washed with a saline solution (0.8 % NaCl), observed alive using an Olympus BX40 microscope, and then fixed in 70 % ethanol. For better identification of internal structures, nematode parasites were cleared in Amann's lactophenol. Parasites were classified based on their morphology and internal structures to the lowest possible taxonomic level and larval stage, following several keys and descriptions [39–41]. Digenean trematodes infecting sardines were identified and quantified, but no lower taxonomic level was specified because the present study focused on nematode parasites.

### 2.5. Data Analysis

Following Bush et al. [42], the prevalence of parasites (P) was calculated as the percentage of sardines infected with a given parasite species, the mean intensity as the average number of parasites in infected fish, and the mean abundance as the average number of parasites found in the sample, including uninfected individuals. We made use of free Quantitative Parasitology 3.0 software [43], which was developed to analyse the aggregated distribution of parasites [44]. In this way, the prevalence (Fisher's exact test), the mean intensity (excluding non-parasitised individuals), and the mean abundance

(including both parasitised and non-parasitised individuals) (Bootstrap t-test) for each parasite species in the different seasons of the years and between coasts were obtained.

R software version 3.5.1. (R Development Core Team, 2018) was used for defining differences in the gonadosomatic index (GSI), the Le Cren's Condition Factor (Kn) and the total fat content values during the year, throughout the reproductive cycle and between thermal regime areas. When normality and homoscedasticity could be fulfilled using Shapiro–Wilk and Levene's tests, respectively, an independent two-sample t-test or one-way analysis of variance (ANOVA) was carried out, followed by a Pairwise t-test. If normality could not be fulfilled, data were analysed using several non-parametric tests (Mann–Whitney U test (U) or a Kruskal–Wallis test), followed by the post-hoc Dunn test with Bonferroni adjustment. On the other hand, a Welch F-test was performed when normality was achieved but not homoscedasticity. Finally, if normality and homoscedasticity could not be achieved, we transformed the data to normal. Spearman's rank correlation coefficient (rs) was performed to assess the possible relationships between nematode abundance and total lipid content. The level of statistical significance adopted was $p < 0.05$.

## 3. Results

### 3.1. Reproductive Cycle and Energetic Condition

A total of 730 females and 645 males were analysed, with no significant differences in reproduction, condition, or parasite infection parameters identified between sexes. Therefore, males and females were grouped in the subsequent analyses. Comparing thermal regime areas, the mean total body length (TL) and total body mass (TW) were not significantly different between them. In relation to condition, a positive correlation (rs = 0.82, $n = 1375$, $p = 0.000$) between the tissue fat content measured by the *Distell Fish Fat Meter* and the visual mesenteric fat stages was observed.

Significant differences were obtained in the monthly relative frequency of gonad developmental phases between the two thermal regime areas in October ($p = 0.007$), December ($p = 0.006$), January ($p = 0.001$), February ($p = 0.000$), August ($p = 0.000$) and September ($p = 0.000$) (Figure 2). Regarding the onset of the spawning season, actively spawning (AS) sardines appeared in October and lasted until March on the northern coast (Figure 2A). By contrast, on the southern coast, AS sardines were found from later in November until April (Figure 2B). The main spawning peak occurred in January in the north (73.3 % AS) and a month later in the south (73 % AS). Individuals in the regressing (RG) stage started to appear progressively, first at a low percentage during February in the north (9 %) and in the south (2 %). Regeneration (RT) of the gonads started in April on both coasts. Sardines in the developing stage DV appeared in September on the northern and southern coasts (10.6 % and 33.6 %, respectively), and were found until December in both areas.

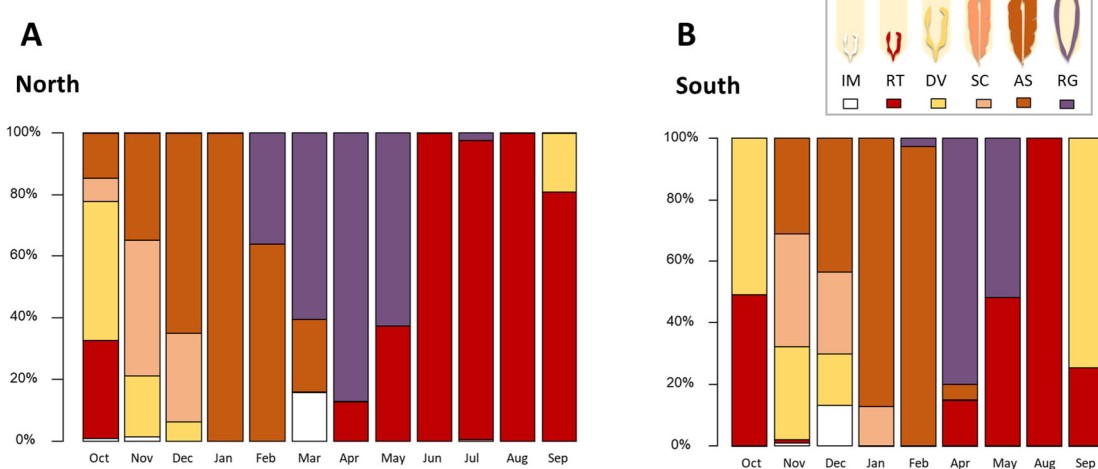

**Figure 2.** (**A,B**) Monthly variations in the relative frequency of gonad developmental phases in sardines captured on the northern coast and in the southern coast, respectively (from October 2019 to July 2021). Along the southern coast, samples could not be collected during March, June and July because of COVID-19 restrictions. Gonad developmental phases are classified into immature (IM), regenerating (RT), developing (DV), spawning capable (SC), actively spawning (AS), and regressing (RG) stages.

No significant differences in the gonadosomatic index (GSI) were obtained in the comparison among stages between areas. As expected, GSI varied significantly throughout the gonad developmental phases ($p = 0.000$) on each coast. Considering all the specimens, higher values of GSI were observed in individuals with spawning capable (SC) (4.65 ± 1.30) and actively spawning (AS) gonads (4.11 ± 1.98), and lower in the regenerating (RT) stage (Figure 3A). The condition varied significantly throughout the gonad developmental phases of the reproductive cycle on both coasts ($p = 0.000$) (Figure 3B). Higher values were identified in the regenerating (RT) (north, 1.06 ± 0.07; south, 1.05 ± 0.07) and developing (DV) (north, 1.05 ± 0.07; south, 1.03 ± 0.07) phases.

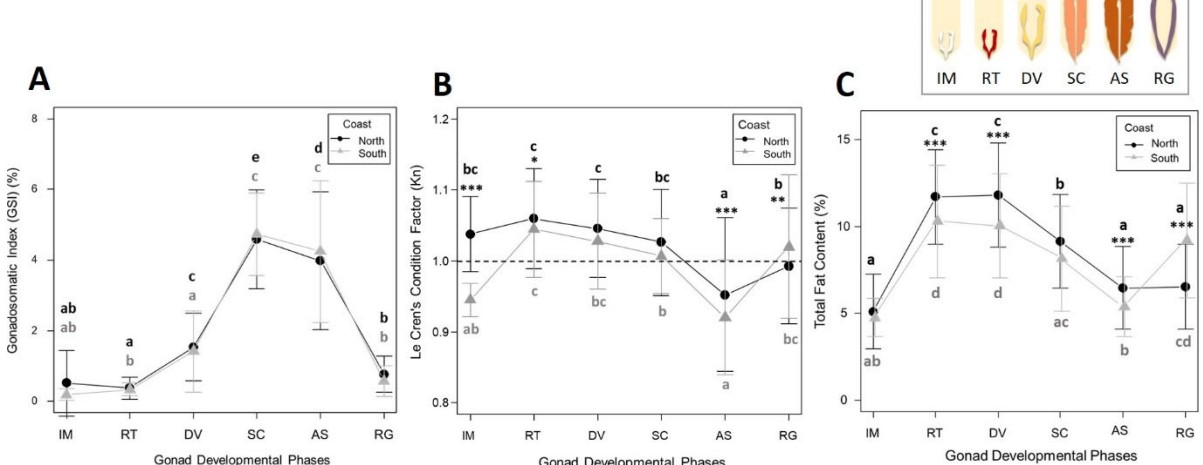

**Figure 3.** (**A–C**) The gonadosomatic index (GSI) (%), Le Cren's condition factor (Kn), and total fat content during the annual reproductive cycle, comparing sardines caught on the northern Catalan Coast with the ones caught on the southern Catalan Coast. Gonad developmental phases: IM, immature; RT, regenerating; DV, developing; SC, spawning capable; AS, actively spawning; RG, regressing. Significance values were indicated as follows: $p < 0.05$ *; $p < 0.001$ **; $p < 0.0001$ ***.

Lowercase letters indicate differences in GSI, Kn and total fat content throughout the gonad developmental phases.

The lowest values of Le Cren condition factor (Kn) appeared in the actively spawning (AS) (0.95 ± 0.11) and regressing (RG) (0.99 ± 0.08) phases of northern sardines, and in AS (0.92 ± 0.08) and immature (IM) (0.95 ± 0.02) specimens caught in the warmer southern area. Comparing coasts, sardines from the northern coast presented significantly better relative condition than those from the south, throughout almost all of the annual reproductive cycle.

Similar to the Kn, the tissue fat content varied significantly throughout the annual reproductive cycle in both sampling locations ($p = 0.000$) (Figure 3C). The worst tissue fat content values were observed in immature (IM) (5.03 ± 1.97), actively spawning (AS) (5.92 ± 2.15), and regressing (RG) (7.27 ± 2.94) individuals, and the best values during the regenerating (RT) (11.36 ± 2.90) and developing (DV) (10.79 ± 3.13) phases. As a general pattern, global data, including parasitised and non-parasitised individuals, showed significantly higher condition in sardines captured in the northern sampling area in almost all reproductive phases (RT, DV, and AS, $p = 0.000$ for each). In contrast, the southern coast presented significantly higher lipid content during the inactive reproductive season (RG) ($p = 0.000$). Lipid content, without considering the presence of parasites (only non-parasitised individuals), presented significant differences between coasts for the same stages, RT ($p = 0.001$), DV ($p = 0.016$), and AS ($p = 0.014$).

### 3.2. Characterisation of Parasitic Infection

A total of 816 nematode larvae were detected in sardines from the Catalan Coast. Of these, 810 larvae belonged to *Hysterothylacium* spp., and six belonged to *Contracaecum* spp. from the family Raphidascaridae and Anisakidae, respectively. No *Anisakis* genus was observed. As reported in Table 2, the overall nematode parasite load, including prevalence, mean intensity and mean abundance, was significantly higher in sardines from the southern coast. During parasite examination, digenean trematodes were found. Regarding this taxon, no significant differences in prevalence were detected between sampling areas.

**Table 2.** Nematodes found in *Sardina pilchardus* individuals in two areas with different thermal regimes in the Catalan Coast, located in the Northwestern Mediterranean Sea, the more northerly Costa Brava and the more southerly Ebro Delta. Hyst, *Hysterothylacium* spp; CA, *Contracaecum* spp; Total, overall nematode parasites; Digenean trematodes.

| | NI | | | Prevalence | | | | | Mean Intensity ± SD | | | Mean abundance ± SD | | |
|---|---|---|---|---|---|---|---|---|---|---|---|---|---|---|
| | Hyst | CA | Total | Digenean | Hyst | CA | Total | Digenean | Hyst | CA | Total | Hyst | CA | Total |
| **North** *n* = 847 | 276 | 4 | 278 | 106 | 32.6 | 0.5 | 32.8 | 12.5 | 1.500 ± 0.851 | 1.000 ± 0.000 | 1.504 ± 0.857 | 0.489 ± 0.854 | 0.005 ± 0.069 | 0.493 ± 0.860 |
| **South** *n* = 528 | 206 | 2 | 207 | 71 | 39.0 | 0.4 | 39.2 | 13.4 | 1.922 ± 1.433 | 1.000 ± 0.000 | 1.923 ± 1.433 | 0.750 ± 1.296 | 0.004 ± 0.061 | 0.754 ± 1.298 |
| *p* | - | - | - | | 0.017 | 1.000 | 0.017 | 0.620 | 0.001 | - | 0.002 | 0.001 | 0.789 | 0.001 |

Differences ($p = 0.000$) in the annual nematode parasite prevalence were observed in sardines distributed along the Catalan Coast (Figure 4). Two different parasitic infection patterns were observed. In the north, sardines captured in winter presented the highest parasite prevalence, followed by sardines sampled in autumn and spring (data in Table 3). In contrast, a higher prevalence in the south was detected in spring, followed by winter and autumn. Nevertheless, summer was the season with lower presence of nematodes for

both locations. By comparing sampling locations, significant differences in parasite prevalence were observed during spring ($p = 0.000$), and no annual variation of parasite mean intensity was detected in sardines from the Catalan Coast. Values shown in Table 3 suggested that sardines were more heavily infected (regarding mean intensity) in autumn-winter and less infected during the summer. Differences between sampling areas in mean intensity and abundance of parasites were detected in winter ($p = 0.029$) and spring ($p = 0.002$), respectively, with significantly higher values in the south.

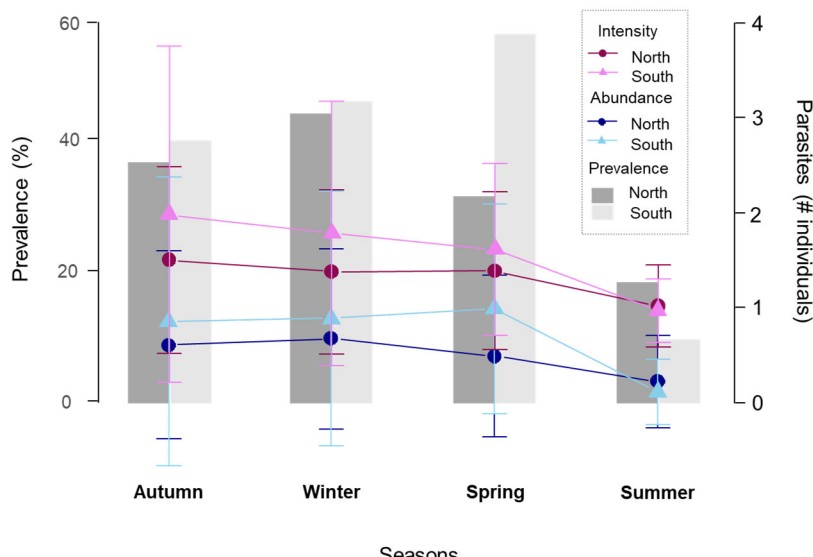

**Figure 4.** Nematode parasite prevalence, mean intensity, and abundance between coasts during the annual seasons.

**Table 3.** Overall parasitic parameters during the annual season for sardines from the northern Catalan Coast and the southern Catalan Coast: N, number of sardines analysed; Ni, number of sardines infected by nematode parasites; Prevalence (%); Mean intensity ± SD; Mean abundance ± SD.

| | N | | Ni | | Prevalence (%) | | | Mean Intensity ± SD | | | Mean Abundance ± SD | | |
|---|---|---|---|---|---|---|---|---|---|---|---|---|---|
| | North | South | North | South | North | South | *p*-value | North | South | *p*-value | North | South | *p*-value |
| **Autumn** | 224 | 175 | 82 | 70 | 36.6 | 40.0 | **0.533** | 1.646 ± 0.986 | 2.129 ± 1.769 | **0.054** | 0.603 ± 0.992 | 0.851 ± 1.528 | **0.066** |
| **Winter** | 241 | 181 | 106 | 83 | 44.0 | 45.9 | **0.767** | 1.519 ± 0.864 | 1.928 ± 1.395 | **0.029** | 0.668 ± 0.948 | 0.884 ± 1.347 | **0.075** |
| **Spring** | 149 | 80 | 47 | 45 | 31.5 | 56.2 | **0.000** | 1.532 ± 0.830 | 1.756 ± 0.908 | **0.216** | 0.483 ± 0.851 | 0.988 ± 1.108 | **0.002** |
| **Summer** | 233 | 92 | 43 | 9 | 18.5 | 9.8 | **0.065** | 1.163 ± 0.433 | 1.111 ± 0.333 | **0.689** | 0.215 ± 0.488 | 0.109 ± 0.346 | **0.036** |
| *p*-value | - | - | - | - | <0.000 | <0.000 | - | **0.303** | **0.232** | - | **0.844** | **0.883** | - |

Meanwhile, a significant negative relationship was observed between the number of nematodes and sardines' somatic condition on both northern and southern coasts (north, rho = −0.98, $p = 0.004$; south, rho = −0.11, $p = 0.016$), and including all samples together (north + south) (rho= −0.11, $p = 0.000$) (Figure 5).

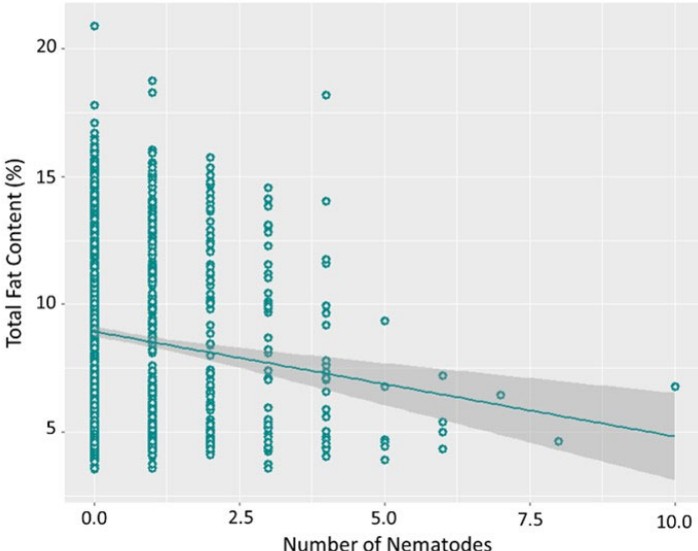

**Figure 5.** Correlation between total fat content (%) and the number of nematodes infecting sardines on the Catalan Coast.

## 4. Discussion

Key biological parameters to evaluate sardine status, such as energetic body condition, reproduction and parasite infection, showed differences between the two analysed thermal scenarios. The only coinciding feature in sardines caught in the two areas was the absence of parasitisation by the genus *Anisakis*, the major agent of anisakiasis in the Mediterranean [45]. After analysing more than 1300 sardines, neither larvae nor adult individuals of this nematode parasite were observed.

*Hysterothylacium* sp. (Rudolphi, 1802) was the most abundant nematode parasite found, showing a higher infection rate than that reported for the NW Mediterranean before 2015 [46,47]. Nevertheless, only a few isolated cases of non-invasive human anisakidosis have been described for *Hysterothylaicum* spp. and *Contracaecum* spp. [48,49], as these nematode parasites infrequently infect fish muscle [50]. Thus, together with the minor or null presence of *Anisakis* spp. in the sardines analysed, it can be confirmed that, in terms of parasites affecting human health, the sardine of the Catalan Coast is an excellent asset for consumption.

However, the implications of parasites in European sardines could have been underestimated. Although the genera *Hysterothylacium* has been reported to produce negative impacts on larvae and adult fish [51,52], most studies have not found a clear relationship between parasite infection and pathogenicity in fish species. The increase of copepods on the Catalan Coast [53] due to global change could be responsible for the higher infection rate by *Hysterothylacium* spp. in sardines from the northwestern Mediterranean, as they become an important intermediate host for this parasite [54] and the most consumed resources in both the larval and adult sardine stages [55]. Therefore, it could be essential to relate the presence of this nematode genus to the body condition of this small pelagic species, to understand sardine health status.

The second major finding of our study was the significant differences in sardine somatic condition, reflected by both the indirect Le Cren Factor (Kn) and the stored lipid content (tissue fat content and mesenteric fat stage), as well as the divergent nematode infection rates, with thermal regimes. The body condition of the analysed specimens varied significantly, following the reproductive cycle with an inverse correlation with the gonadosomatic index (GSI), as expected, due to its capital breeder strategy [53]. However, our results showed significant differences along latitude, with clear differentiation in condition between sardines from the cooler north and the warmer south. Northern sardines presented higher Kn values and were generally in better condition throughout

the reproductive cycle. In line with Kn values, the sardines' condition reflected by stored lipid content was also clearly higher on the northern coast, during almost all of its reproductive cycle. These differences can be attributed to several biotic and abiotic factors (e.g., seawater temperature and primary productivity).

A similar pattern of significant differences between north and south (cold vs. temperate waters) was also observed in the prevalence, intensity and abundance of parasites. Parasite infection could be influenced by physical systems effects on the availability of prey and transference of nematodes. Higher levels of parasite infection were reported in fishes caught along the southern, warmer and more stratified areas, perhaps due to differences in the availability of intermediate hosts [54,56,57]. Moreover, higher temperatures favour the development and the reproductive cycle of *Hysterothylacium aduncum* [58]. The present study highlights significant seasonal differences in parasite load, with lower prevalence during the summer and higher in winter-spring. Excluding the summer period (in which low and similar nematode prevalence were found between coasts), a higher prevalence trend has been observed in the south, although significant differences were found only during spring. In winter, when sardine have translocated all energy reserves in the development of their gonads, individuals showed high and similar prevalence between thermal regimes. The same trend was observed for parasite intensity, which was low and similar between coasts during the summer, but significantly higher in the southern area during the winter.

After analysing body condition of non-parasitised specimens between coasts, differences in tissue fat content were observed between thermal regimes, which indicates that there are other factors involved in affecting sardine condition. However, nematode abundance may represent a potential factor to be considered in body condition decrease. Regarding the relationships among the studied biological parameters, this work revealed a significant negative correlation between parasite abundance and fish condition. Nevertheless, we should be cautious with this statement, as there was a low number of specimens with high nematode abundance, as indicated in Figure 5. A question to be answered is whether higher parasite infection causes a lower condition or, on the contrary, sardines with a lower condition are more prone to higher load of parasites. The low prevalence and intensity during the summer coincides with a better fish condition, which may favour overcoming parasite infection. In contrast, during the winter, a higher parasite load coincides with a lower somatic condition, which may hinder the overcoming of parasite infection. Moreover, it has been hypothesised that the high prevalence and intensity of infection in winter may be related to the lack of mesenteric fat surrounding visceral organs, which could accelerate the perforation of the organ wall tissues, going towards the visceral cavity and remaining stored.

Regarding spawning dynamics, it is known that the main spawning peak of the European sardine occurs during the coldest and most oligotrophic months (e.g., [26]; among others). However, the present study reported significant differences in the onset of the spawning season and the main spawning peak in the northwestern Mediterranean. The spawning season on the northern coast started about one month earlier (in October), that in the south (November). Similarly, a one month delay in the main spawning peak was also observed in the south (February) compared to the north (January). In fact, some studies have highlighted variations in reproductive strategy, timing of the onset and cessation of the spawning season, fecundity and egg quality, mainly linked to the life history traits of the stock shaped by the environment over time [14,18]. Sardines in the regenerating (RT) and developing (DV) gonadal stages had the best condition along the whole sampling area, coinciding with the late winter-early spring phytoplankton bloom, linked to the increasing temperature [59–61]. This process provides a direct food intake to sardines, which are starting to recover from the spawning season [62,63]. Hence, the European sardine feeds intensively during the maximum primary production month, during the reproductive resting phase, and acquires sufficient energy reserves including lipid in muscle and mesenteric fat [9]. It should be pointed out that, during the regressing

stage (RG) in spring, a significant higher energetic body condition was observed in the south.

These differences found between the northern and southern sardines regarding condition, reproduction, and parasite infection have been attributed to superficial and intermediate sea temperature (SST), as the Ebro Delta (in the south) and surrounding waters are significantly warmer (1–2 °C) compared with the Costa Brava (in the north) during the year, potentially affecting body condition due to sardine's cold–temperate water preference [27,64]. Moreover, mechanisms such as strong winds, frontal and sub-mesoscale processes and the confluence of deep ocean convection in the close Gulf of Lion, which produce water mixing and fertilization in the northern coast, induce higher nutrient levels and a delayed strengthened bloom, creating a productive feeding ground for sardine [15,65]. The Rhone and Ebro riverine inputs in the Catalan Coast are advected by the Northern Currents and produce high values of nutrients and chlorophyll-a in the photic layers [65,66]. The sea temperature rise and extreme temperature phenomena (e.g., in summer 2022, the Catalan Coast registered the highest SST, with records of more than 28 °C) promoted by climate change, could influence the current dynamics and prolong the stratification period, leading to more oligotrophic waters [67]; it can be difficult for these to facilitate intensive feeding, to allow recovery after spawning. These abiotic mechanisms influence primary productivity [68,69] and cause changes in the abundance, health status and spawning dynamics of small pelagic fish, with direct consequences for recruitment strength [70]. On the Catalan Coast, a primary production bloom southwards has previously been reported [25], resulting in an earlier food intake and rapid condition enhancement in sardines from this area [26]. This may explain why the southern sardines show better condition during the regressing phase, in contrast to what happens for the rest of the year.

Relating seasonal parasitisation patterns to diet, it should be mentioned that, during winter-spring, the prevalence and intensity of nematodes was higher, perhaps due to the outcrop of nutrients [24] producing higher zooplankton biomass, which acts as intermediate hosts of the nematodes. During the stratification period, the Ebro riverine runoff maintains a high primary production in the surface near the outfall [71], which could be an explanation for the significantly higher prevalence and mean abundance of nematodes in southern sardines during spring. As stated before, sardine feed intensively during the summer, in order to devote reserves to reproduction in the coldest months of the year. In this context, another hypothesis is that a stronger feeding activity would hinder the accumulation of parasites in the stomach of sardines during this season. During the spawning season (in winter), sardine continue feeding at low levels and change from filter to particulate feeding [72], implying an annual shift in diet [55,73], which may entail the ingestion of more parasitised plankton.

Present and projected rising temperatures in the area may lead to an increase in the parasitic load and decrease in the general condition of European sardines. Thus, current southern reproduction features, condition, and parasite infection will be potentially homogenised towards the north, becoming even more pronounced in the meridional latitude.

## 5. Conclusions

The present study provides an important global view of the annual body condition of sardine, directly related to the reproductive cycle and strongly influenced by environmental mechanisms such as seawater temperature and primary production rates, as well as parasite incidence. Significantly better condition was recorded in sardines sampled on the colder northern Catalan Coast. Regarding parasite infection between thermal regimes, significant differences were observed in parasite load, with higher nematode infection in the warmer southern area. The absence of *Anisakis* spp. infection in sardines from the Catalan Coast is of particular interest. This study gives detailed information on the annual variation of parasites. Overall, it highlights the nematode

parasite *Hysterothylacium* spp. as the main nematode infecting sardines, with no pathogenicity for human consumption. Nematode load may stand out as a potential factor involved in the reduction of sardine condition, since a negative correlation was obtained between parasite abundance and body condition, although it remains to be elucidated whether this relationship is one of cause or consequence. This finding could become of major concern if parasite infection increases, due to a consistent warming pattern in the Mediterranean Sea.

**Author Contributions:** X.F.-T.: Conceptualization, Formal analysis, Investigation, Writing, Visualization; M.C.-H.: Investigation, Revision; J.V.: Conceptualization, Revision, Supervision, Project administration; M.M.: Conceptualization, Revision, Supervision, Project administration. All authors have read and agreed to the published version of the manuscript.

**Funding:** This study was financially supported by the European Maritime and Fisheries Fund (EMFF) 2019–2021; Ref; 152CAT00012, ARP059/19/00011.

**Institutional Review Board Statement:** Ethical review and approval were not necessary since the fish used in this study were dead and obtained from the fishing sector, destined for human consumption.

**Data Availability Statement:** Sea surface temperature (SST, °C) data are openly accessible in the HadISST sea surface temperature time series (1900–2020) (https://www.metoffice.gov.uk/hadobs/hadisst/ (accessed on 22 February 2022)). Data on condition, reproduction, and parasites is available upon request to the authors.

**Acknowledgments:** We want to thank the fishermen's guild for providing sardine samples. Our gratitude goes to Lluís Rodríguez, captain of the vessel Agneta (Palamós) for valuable advice and major interest in the project.

**Conflicts of Interests:** The authors have no conflicts of interest to declare.

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
