# Peer review of "Influence of Thermal Regimes on the Relationship between Parasitic Load and Body Condition in European Sardine along the Catalan Coast"

_fishes, doi:10.3390/fishes7060358_

Round 1

Reviewer 1 Report

General comments:

The European sardine is an economically important clupeid fish with planktivorous habits, and the stocks are affected by global warming. To evaluate effects of increasing water temperature on sardine health status, fish body condition and parasite infection were compared between the northern coast and the southern coast along the Catalan Coast. Significant higher Le Cren condition and fat content were detected in the northern coast, whereas higher prevalence, mean intensity and mean abundance were found in the southern coast. Also there was a significant negative correlation between parasite abundance and fish somatic condition both in the northern and southern coasts. These findings suggested there was worse fish body condition and higher parasite infection in high temperature areas, which are common and reasonable. Thus it is plausible that parasite infection would increase under the global warming. Relative to the northern coast, however, the bad fish condition  may do not result from the high nematode infection in the southern coast, but other factors, such as water temperature and gonad development. The manuscript should be reorganized to avoid to be misunderstood. The specialized words on Parasitology also need to be improved. So this manuscript may be accepted after major revision. 

Specific comments:

Title:

Emphasis of the research is the differences of nematode infection and body condition in sardine in a southern and a northern coast, but not the effect of nematode on body condition. Therefore the title should be changed to avoid causing ambiguity. 

Abstract:

-Line 14-16 What’s the “affect” mean? Decrease or increase the primary productivity and then fish condition? Parasitism is a type of symbiotic relationship or phenomenon. It should be replaced with parasite or nematode here and in the whole manuscript.

-Line 19-21 Specific results on reproductive cycle and fish condition should be provided here, which are very important.

-Line 24 Not only the prevalence but also the mean intensity and mean abundance showed seasonal changes.

-Line 27-27 This sentence should be rewritten. Primary productivity is not abiotic factor. Perhaps “differentiation in nematode infection and fish condition along the annual cycle….” 

Introduction:

-Line 69 replace parasitism with parasite infection. 

Materials and methods:

-Line 135 2.4 Parasitological examination

-Line 146-150 It is better to move these analysis to the next part. Change “unaffected” to “uninfected”.

-Line 158-160 To rule out the possibility of effect of the parasite on fish condition, it is necessary to compare the body condition of the uninfected fish between the two coasts in the whole year and along the different seasons.  

Results:

-Line 198-200 Differences in GSI between both areas should be tested not only in the whole year, but also along the different seasons, owing to the lagging gonad development in the northern coast as mentioned in the previous paragraph. 

-Line 191-196 Leave the highest identity and move the other sequence identity results to Table 3 on the upper right.

-Line 209 It is another topic from “In the present…”. So a new paragraph should begin from here.

-Line 221-235 Explanations on OIM and PIM are not the emphasis of this manuscript, but the novel species identification. So please discuss this with few sentences. It is also necessary to add some references to support the discussions.

-Line 229-234 Although significant higher mean abundance was detected in the south, but with low mean abundance in both areas, such as 0.493 in the north and 0.754 in the south. Therefore, the nematode infection may affect host body condition, but it is not convincing that the differences in fish condition between the two areas are attributed to the nematode infection.

-Line 259 It is not mean intensity but abundance or number of nematode. Despite of the negative correlation, the correlation degree is low with -0.11. The result may be due to the few  the specimen number in the high abundance group as indicated in Figure 5, such as only one sample in 7, 8 and 10 abundance group, respectively.  

Discussion:

-Line 281 “Previous reported parasitic load by theses nematodes were lower”, where did the results come from? Higher mean abundance of Hysterothylacium sp.  was reported in the European sardine in different season by Fuentes et al., 2022(Nematode Parasites of the European Pilchard, Sardina pilchardus (Walbaum, 1792): A Genuine Human Hazard. Animals, 2022)

-Line284-285 Fish condition was affected by nematode abundance, but not nematode prevalence. “This high nematode prevalence” is not consistent with the result with low mean abundance.

-Line 378-385 Regarding the effect of nematode infection on fish condition have been discussed in the third paragraph in Discussion part. Please move this paragraph ahead.

-Line 387-396 This is the summary of the manuscript. Please merge it with the Conclusion part. 

Conclusion:

-Line 403 Remove the word “rates” due to infection rate means prevalence.

-Line 409 Not parasite intensity but abundance as showed in Figure 5.

Reviewer 2 Report

Dear authors,

First, I want to congratulate you on the excellent work presented in this manuscript. It is well written and has original and novel concepts regarding the parasitism incidence on the sardines condition, which can be extremely useful for stock management on this species.

I only have minor spelling corrections/suggestions to make about your manuscript, which you can check on the attached document. Please take into consideration my suggestion for your manuscript title.

I have only two more suggestions to improve your manuscript readability:

- Insert a table in the Material and Methods 2.1 section that specifies the number of fish sampled per thermal zona in each month, to make clear that the samples are similar and representative;

- Insert an Image at the final of the Discussion section that can sum up all your findings (please see the note on the attached document).

Best regards,
